

# A survey of streaming data anomaly detection in network security

Pengju Zhou

Cyber Science and Engineering, Sichuan University, Chengdu, Sichuan, China

## ABSTRACT

Cybersecurity has always been a subject of great concern, and anomaly detection has gained increasing attention due to its ability to detect novel attacks. However, network anomaly detection faces significant challenges when dealing with massive traffic, logs, and other forms of streaming data. This article provides a comprehensive review and a multi-faceted analysis of recent algorithms for anomaly detection in network security. It systematically categorizes and elucidates the various types of datasets, measurement techniques, detection algorithms, and output results of streaming data. Furthermore, the review critically compares network security application scenarios and problem-solving capabilities of streaming data anomaly detection methods. Building on this analysis, the study identifies and delineates promising future research directions. This article endeavors to achieve rapid and efficient detection of streaming data, thereby providing better security for network operations. This research is highly significant in addressing the challenges and difficulties of analyzing anomalies in streaming data. It also serves as a valuable reference for further development in the field of network security. It is anticipated that this comprehensive review will serve as a valuable resource for security researchers in their future investigations within network security.

## INTRODUCTION

In recent years, with the rapid increase in the number of network devices, the generation of streaming data from these devices has catalyzed considerable interest in real-time processing. A significant amount of research work has been dedicated to the formulation of efficient anomaly detection solutions. Currently, anomaly detection for streaming data has been applied to various scenarios, such as network intrusion detection (*Tidjon, Frappier & Mammar, 2019*; *Wahab, 2022*), fault detection (*Bagozi, Bianchini & De Antonellis, 2021*; *Bonvini et al., 2014*), medical diagnosis (*Ren, Ye & Li, 2017*; *Podder et al., 2023*), fraud detection (*Laleh & Abdollahi Azgomi, 2010*; *Dal Pozzolo et al., 2015*), and network flow analysis (*Pramanik et al., 2022*; *Tang et al., 2020*). Streaming data in the field of cybersecurity primarily originates from system logs, network traffic, intrusion detection devices, and other sources of high-speed, real-time data. These data contain features related to network anomaly events, such as IP addresses and protocol types. While conventional network anomaly detection methods demonstrate commendable performance on static or offline data. However, when faced with streaming data, the

Corresponding author
Pengju Zhou,
zhoupengju@stu.scu.edu.cn

inability to perform online learning and real-time model updates poses significant challenges for network security that requires real-time anomaly detection and feedback. In response, researchers have introduced various solutions to address these challenges. Some reviews furnish a broad survey of streaming data anomaly detection algorithms aiming to help understand related concepts, history, and methods. This article comprehensively reviews recent articles on streaming data anomaly detection in cybersecurity and categorizes the scenarios, data types, research methods, and problems addressed. This research aims to achieve two objectives:

- Systematically categorize existing research to provide network security researchers with a clear understanding of the current state of the field.
- Summarize several future research directions based on the classification results to help researchers quickly focus on these directions.

The research structure is as follows: "Research Background" introduces common concepts in anomaly detection. "Articles Selection for Literature Review" describes a rigorous systematic review methodology that includes comprehensive multi-database search strategies, explicit inclusion criteria, and a three-stage filtering process. "Challenges and Requirements in Streaming Data for Anomaly Detection" discusses the challenges and requirements faced by streaming data anomaly detection. In "Proposed Classification Method", this article categorizes anomaly detection datasets, measurement techniques, detection algorithms, result types, reviews and compares related algorithms. "Study of Literature and Discussions" provides a detailed study and discussion of the literature surveyed. "Future Directions and Open Research Challenges" proposes some new research ideas. Finally, "Conclusions" presents the conclusions and provides clear guidance for further work.

## RESEARCH BACKGROUND

In order to fully understand the research content of streaming data anomaly detection, it is imperative to first elucidate several fundamental concepts. These concepts form the foundational framework for understanding subsequent discussions and furnish the necessary theoretical background for an in-depth exploration of algorithms for streaming data anomaly detection. The following is a detailed explanation of these pivotal concepts:

**Anomaly detection:** It can be traced back to early statistics and refers to situations that do not match other patterns (*Grubbs, 1969*). Currently, anomaly detection is generally understood as the identification of events that are uncertain and do not conform to expected patterns.

**Streaming data:** Streaming data refers to a potentially infinite sequence of data items that arrive continuously at a fast pace. It is defined as $\{(x_t, y_t)\}_1^\infty$, $(x_t, y_t)$ represents a data item that arrives at time $t$. $x_t \in R_n$ is a $n$-dimensional feature vector, $y_t \in Y = \{c_1, c_2, \ldots, c_k\}$ and is the class label associated with the data item.

**Concept drift:** Concept drift was proposed by *Schlimmer & Granger (1986)*. Formally, the process generating the streaming data can be considered as a joint distribution over random variables $Y$ and $X = \{X_1, X_2, \ldots, X_n\}$, where $y \in \text{dom}(Y)$, $Y$ and $X = \{X_1, X_2, \ldots, X_n\}$, where $y \in \text{dom}(Y)$ represents the class label and $x_i \in \text{dom}(X)$ represents the attribute values, with $\text{dom}(\cdot)$ denoting the domain of a random variable. The concept at time $t$ can be represented as $P(Y|X_t)$, where $X_t$ refers to the input data. Concept drift occurs when there is a change in the relationship between input data and the target label data due to changes in the characteristics of the data from different sources at different time $t$, $P(Y|X_{t1}) \neq P(Y|X_{t2})$. The various types of concept drift encompass sudden concept drift, gradual concept drift, incremental concept drift, and recurring concept drift (*Ramírez-Gallego et al., 2017*).

**Feature drift:** Let $F$ denote feature space at $t$, where $F * t \subseteq F$ represents the highest discriminative subset of features. Feature drift occurs when there is a change in the feature subset, $F * t_i \neq F * t_j, t_i \neq t_j$. Feature drift can occur both when there is a change in the data distribution or when there is no change (*Barddal et al., 2017*).

**Concept evolution:** Let $Y = \{c_1, c_2, \ldots, c_k\}$ be a set of classes from the training set used to train a classifier, representing the known concepts related to the underlying problem. The classes in $Y$ are referred to as known or existing classes. During the generation of the streaming data, a new class emerges that does not exist in $Y$. This class is referred to as an emerging class, and this phenomenon is known as concept evolution (*Kulesza et al., 2014*).

**Time window:** Time window refers to a subsequence between the $i$th and $j$th arrivals in a sequence, denoted as $w[i, j] = (x_i, x_{i+1}, \ldots, x_j)$, where $i < j$. Time windows can be categorized as Landmark window, Time-dampened window (also known as Fading window), Sliding window, and Tilted-time window (*Nguyen, Woon & Ng, 2015*). Figure 1 for a visual representation of these categories.

# ARTICLES SELECTION FOR LITERATURE REVIEW

## Overview of the review process

This investigation utilized a systematic methodology to select high-quality articles on streaming data anomaly detection, with pronounced emphasis on network security applications. It established a structured review framework comprising: (1) comprehensive search across multiple databases, (2) the application of stringent inclusion criteria, and (3) the meticulous filtering and critical appraisal of the resultant literature.

## Data sources

The systematic review was conducted from September 2023 to October 2024, focusing on publications from 2015–2024. The present study conducted an exhaustive and detailed search of relevant literature, including research articles from journals, conferences, books, and magazines. To guarantee comprehensive coverage, multiple databases were utilized, including Google Scholar, IEEE Xplore, Springer, ScienceDirect, Scopus, Web of Science, and ACM Digital Library. Furthermore, a selection of highly reputable conferences such as

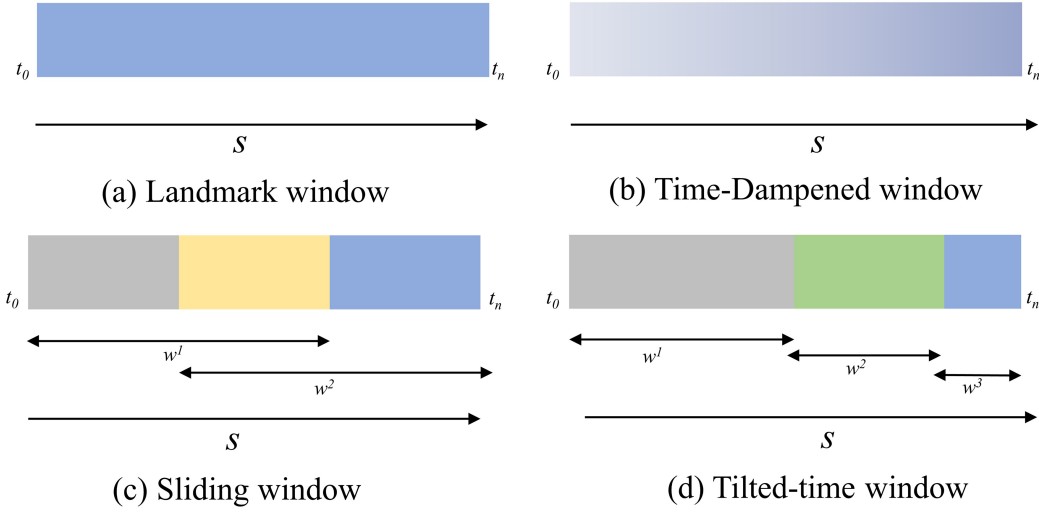

**Figure 1 Time window model.** (A) Landmark window. (B) Time-dampened window. (C) Sliding window. (D) Tilted-time window.

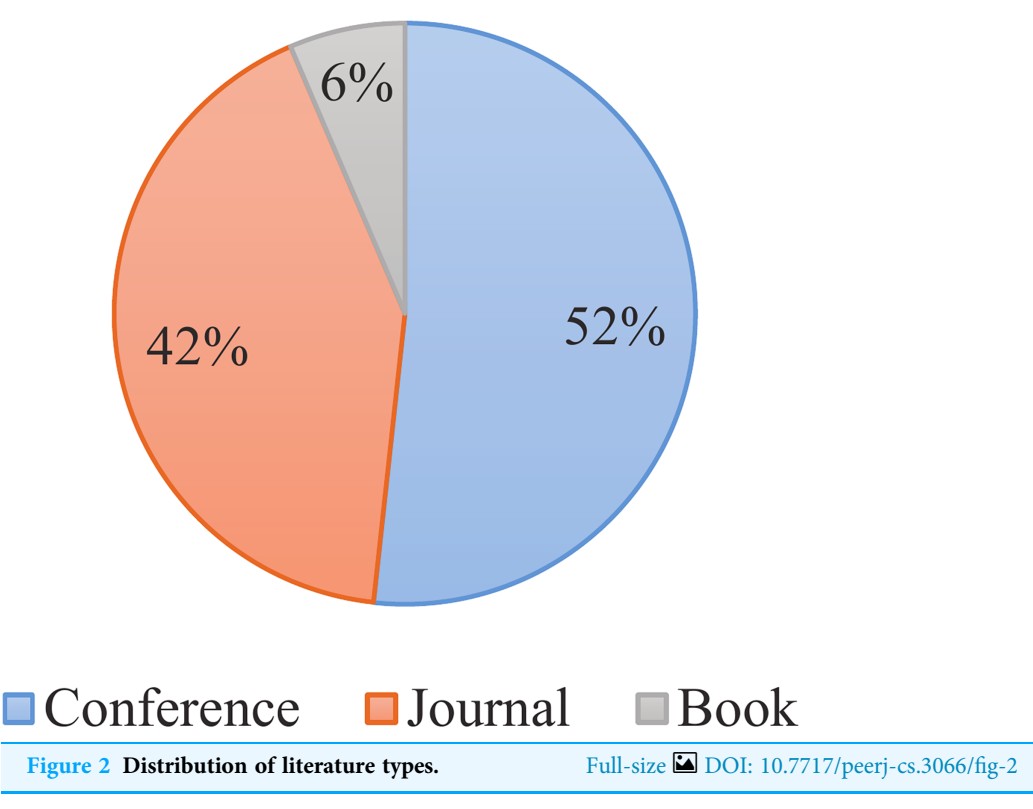

**Figure 2 Distribution of literature types.**

USENIX, WWW, SIGKDD, VLDB, SIGMOD, NDSS, S&P, ICDM, and CCS were considered. Figure 2 visually presents the distribution of research from different resource types. As depicted, the *corpus* of literature under review comprises 884 conference articles, 715 journal articles, and 110 books included in the study.

## Search keywords and scope

The study search was confined to computer science and engineering domains, utilizing a two-phase keyword strategy:

- Initial phase: (Stream* data OR Stream* OR Real-time data) AND (Anomaly detection OR Outlier detect* OR Deviation detection OR Novelty detection).
- Refinement phase: Added domain-specific terms (IoT OR ICS OR SDN OR GRID OR WSN OR Cloud OR Sensor OR Embedded Systems OR Intrusion Detection).

  where "*" indicates wildcard matching to capture variations of the terms.

## Selection criteria

To ensure the quality and relevance of the selected literature, the study established specific inclusion criteria as delineated in Table 1. These criteria guided both initial screening and detailed evaluation phases.

## Filtering process

The study employed a systematic selection process, illustrated in Fig. 3, encompassing three principal stages:

1. Initial screening: In the initial phase, it used the keywords mentioned in the first strategy to screen out a total of 2,620 publications. Subsequently, based on the crude detection in the secondary strategy's criteria, it screened out 909 publications, leaving 1,711 publications. From 1,711 identified articles, this study removed publications lacking academic rankings and those irrelevant based on abstract review, yielding 252 publications.
2. Quality assessment: Further evaluation based on methodological rigor, citation impact, and relevance to streaming anomaly detection reduced the selection to 105 high-quality articles.
3. Final classification: Detailed examination identified 42 articles specifically focused on network security applications of streaming anomaly detection, which formed the core of the analytical review.

## CHALLENGES AND REQUIREMENTS IN STREAMING DATA FOR ANOMALY DETECTION

Streaming data is characterized by real-time, high-speed, infinite, and variable properties. This study delineates the specific challenges and requirements faced by anomaly detection in streaming data in the field of network security.

- Single pass scanning: Owing to inherent storage constraints, detection algorithms should be able to process data in a single pass, strictly adhering to the sequential arrival order of the streaming data.

**Table 1 Criteria for literature selection and explanation.**

| Selection criteria | Explanation |
| --- | --- |
| Journal, conference, book | Comprehensively search across various sources to ensure a broad collection of information. |
| Renowned conference or journal | Prioritize sources from well-regarded platforms to maintain the quality and credibility of the data. |
| High citation or download count | Consider articles with high citation or download counts as these are indicative of the work's recognition and impact within the academic community. |
| Literature from the past 10 years | Focus on the most recent studies to track the latest developments and current trends. |
| Articles proposing detection methods | Specifically look for articles that propose new or improved detection methods to understand the principles and effectiveness of these approaches. |

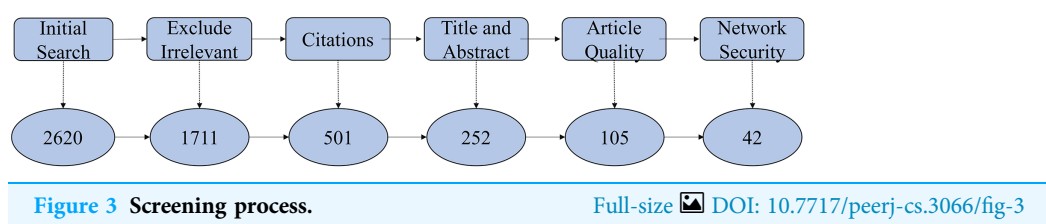

**Figure 3 Screening process.**

- Limited memory: Streaming data is continuous and inexhaustible, making it impractical to retain the entirety of memory. Detection algorithms must use smaller memory space to detect anomalies, ensuring that other applications of the system are not affected.
- Dynamic adaptation: As streaming data evolves over time, detection algorithms must be engineered for dynamic adaptation or incremental updates to maintain high accuracy in a perpetually changing data landscape.
- Result approximation: Traditional anomaly detection algorithms based on static data yield reasonably accurate results. However, streaming data detection requires processing data in extremely short timescales, often necessitates the acceptance of approximate outcomes. Anomaly detection algorithms must therefore strike a delicate balance between computational performance and detection accuracy in this time-constrained environment.

## PROPOSED CLASSIFICATION METHOD

This section presents a systematic classification of methodologies for streaming data anomaly detection. Initially, it delves into the dataset typologies employed in the field of network security anomaly detection, which is pivotal in enhancing the caliber of investigative results. Subsequently, the discourse transitions to the examination of measurement techniques tailored for streaming data, which are instrumental for mitigating storage overhead and accelerating processing throughput. The analysis then proceeds to scrutinize the diverse spectrum of detection algorithms, with a particular emphasis on enhancing their efficacy and precision. The advocated classification schema is presented in Fig. 4.

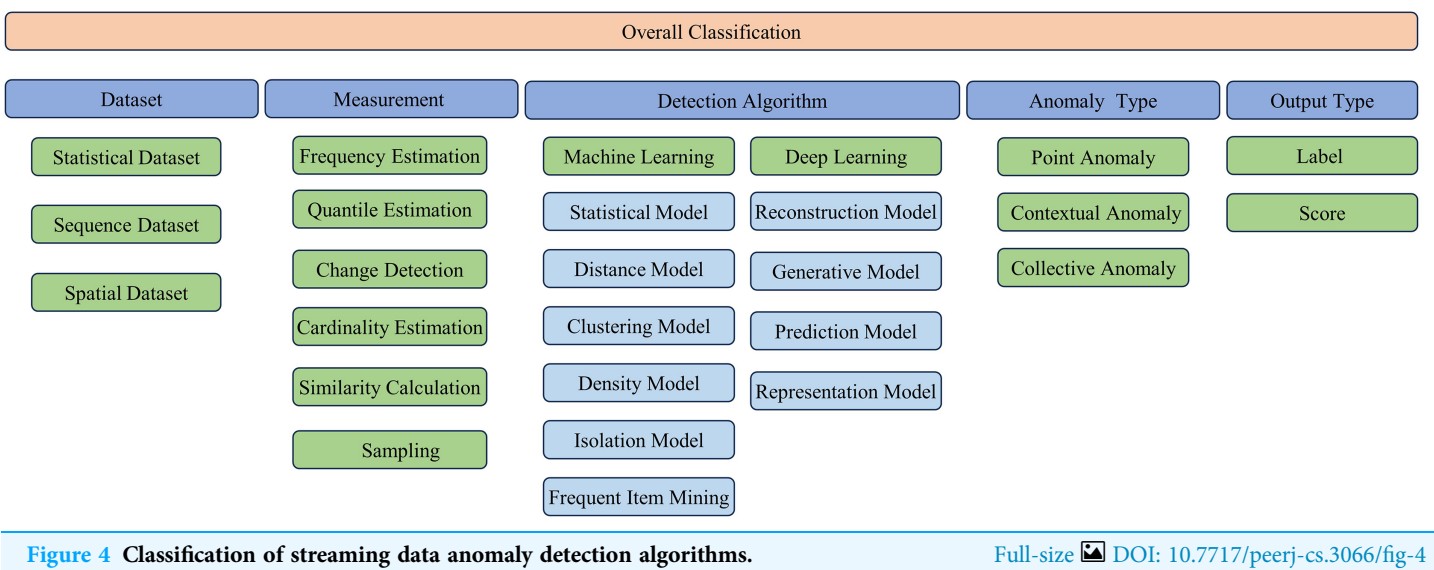

**Figure 4  Classification of streaming data anomaly detection algorithms.**

## Dataset classification

Given that real-world datasets are not universally designed for streaming data, recent research has principally adopted two methodologies. The initial approach identifies anomalies by classifying infrequent occurrences as statistical outliers. The second strategy, congruent with the real-time characteristics of streaming data, pinpointing anomalies that conform to the protocol's real-time criteria. The classification of the datasets is as follows:

- Statistical dataset: It refers to datasets composed of multiple continuous or discrete features presented in tabular form *Moustafa, Turnbull & Choo (2018)*.
- Sequential dataset: It refers to datasets where there is implicit time correlation or dependency between consecutive data points (*Karim, Majumdar & Darabi, 2020*).
- Spatial dataset: It refers to datasets with specific locations and spatial relationships (*Wang et al., 2020a*).

This study presents a compilation of commonly used datasets, with information including the dataset name, data entries, classified attributes, the number of attacks, and the statistically derived anomaly thresholds, as shown in Table 2.

## Measurement technique

Streaming data measurements in large-scale, high-speed networks is crucial for anomaly detection. A multitude of algorithms estimate streaming data for anomaly detection using approximate computation and compression techniques under the stringent constraints of limited memory and computational resources. This study provides a systematic summarization of prevalent streaming data measurement techniques as follows:

- Frequency estimation: Frequency estimation estimates the frequency of elements in streaming data using hash functions and counters. Prominent examples include space-saving, count-min sketch (CMS). CMS uses multiple hash functions and a 2D

**Table 2 Dataset information.**

| Dataset | Records | Attributes | Attack records | Reference |
|---|---|---|---|---|
| KDDCUP 99 | 567,497 | 4 | 3,377 | *The UCI KDD Archive (1999)* |
| CIC 2017 (Thursday-WebAttacks) | 1,070,367 | 4 | 2,180 | *Sharafaldin, Lashkari & Ghorbani (2018)* |
| CIC 2017 (Tuesday-Working Hours) | 445,909 | 3 | 13,835 | *Sharafaldin, Lashkari & Ghorbani (2018)* |
| CIC 2017 (Friday-Morning) | 191,033 | 2 | 1,966 | *Sharafaldin, Lashkari & Ghorbani (2018)* |
| CIC 2017 (Friday-Afternoon-DDos) | 225,745 | 2 | 128,027 | *Sharafaldin, Lashkari & Ghorbani (2018)* |
| CIC 2017 (Friday-Afternoon-PortScan) | 286,467 | 2 | 158,930 | *Sharafaldin, Lashkari & Ghorbani (2018)* |
| CIC 2017 (Wednesday-working Hours) | 692,703 | 6 | 252,672 | *Sharafaldin, Lashkari & Ghorbani (2018)* |
| CTU-13 (48) | 114,077 | 52 | 63 | *Garcia et al. (2014)* |
| CTU-13 (52) | 107,251 | 58 | 8,164 | *Garcia et al. (2014)* |
| Edge-IIoTset | 20,952,648 | 15 | 9,728,708 | *Ferrag et al. (2022)* |
| UNSW-NB15 | 82,332 | 9 | 45,332 | *Moustafa & Slay (2015)* |
| MQTTset | 8,456,823 | 6 | 115,822 | *Vaccari et al. (2020)* |
| CIDDS001 (Week1) | 8,451,520 | 4 | 1,440,623 | *Ring et al. (2017)* |
| CIDDS001 (Week2) | 10,310,733 | 4 | 1,795,404 | *Ring et al. (2017)* |

array to estimate frequencies (*Cormode & Muthukrishnan, 2005*). Given its exceptional space efficiency, rapid query performance, and robust fault tolerance, this approach is widely adopted across diverse applications. *Tong & Prasanna (2017)* employed CMS and *K*-ary sketch for heavy hitter detection and heavy change detection. *Bhatia et al. (2022)* used CMS for detecting anomaly micro-clusters in streaming data.

- Quantile estimation: Quantile estimation achieves quantile estimation by maintaining quantile summaries or quantile estimators within error boundaries, such as *t*-digest (*Radke et al., 2018*) and the Greenwald-Khanna algorithm (*Lall, 2015*).

- Change detection: Change detection identifies change points or anomalies that deviate from the normal pattern by analyzing patterns, trends, or statistical features in streaming data. Among the most commonly employed algorithms are the cumulative sum (CUSUM) control chart and the Page-Hinkley (PH) test. *Martínez-Rego et al. (2015)* adopted Bernoulli CUSUM for change detection. *Duarte, Gama & Bifet (2016)* availed of PH to detect changes in the data generation process and respond to them using pruning rules.

- Cardinality estimation: Cardinality estimation makes use of hash functions or bit arrays to map elements to specific positions and estimate cardinality based on statistical information. In contrast to frequency estimation, cardinality estimation is exclusively focused on the total number of distinct elements in streaming data, such as LogLog, HyperLogLog, and bloom filter. It is worth noting that HyperLogLog estimates cardinality based on the maximum leading zero count (LZC) in an array, while LogLog estimates cardinality based on the maximum zero count in an array. *Xiao et al. (2023)* proposed three HyperLogLog-based algorithms to estimate streaming distribution and reduce estimation errors.

- Similarity calculation: Similarity calculation is the relationship between data objects by comparing their similarities. Locality sensitive hashing (LSH) is a prevalently used algorithm. LSH functions by selecting a family of hash functions engineered to map similar data points to the same bucket with a high probability. Notable variants include MinHash, LSH Forest, and random projection (RP). *Zeng et al. (2023b)* used double locality sensitive hashing bloom filter (DLSH) to improve accuracy and efficiency. *Yang et al. (2022)* proposed DLSHiForest to address the inherent property of infinite, correlated, and concept drift in traditional static anomaly detection algorithms. *Pham et al. (2014)* applied RP to obtain compressed data and solve the scalability issue. *Lai et al. (2022)* replaced the entropy estimation calculation with a simple lookup process using RP.

- Sampling: Sampling algorithms approximate the analysis of entire streaming data by selecting a subset of elements from the streaming data. These include sticky sampling (SS) and reservoir sampling (RS). SS prioritizes sampling based on data priority, while RS is a random sampling technique. *Wang et al. (2023)* adopted weighted RS to model the distribution characteristics of historical reliability streaming data in mobile edge computing (MEC). *Yu et al. (2018)* applied RS algorithm to represent the vectors of vertices in dynamic network computation.

## Detection algorithm

Research on streaming data anomaly detection has evolved through multiple scholarly trajectories. Comprehensive studies have systematically examined diverse detection methodologies, with each addressing distinct aspects: *Wang, Bah & Hammad (2019)* taxonomized approaches into distance-based, clustering-based, density-based, ensemble-based, and learning-based categories (2000–2019); *Boukerche, Zheng & Alfandi (2020)* analyzed algorithmic efficiency parameters and high-dimensional processing challenges while proffering a novel classification framework; *Din et al. (2021)* specifically addressed concept evolution phenomena in streaming classification; *Bhaya & Alasadi (2016)* evaluated streaming mining techniques for network traffic anomaly detection and *Souiden, Brahmi & Toumi (2016)* furnished comparative assessment frameworks for algorithm selection across various contexts. Further specialized research has emerged along two distinct lines. The first focuses on specific application scenarios: *Fahy, Yang & Gongora (2022)* addressed the label scarcity problem in dynamic streams with concept drift; *Stahmann & Rieger (2021)* investigated anomaly detection requirements in Industry 4.0 manufacturing environments with millisecond-frequency sensor data. The second line explores specialized methodological frameworks: *Krawczyk et al. (2017)* examined ensemble learning approaches for non-stationary stream environment; *Faria et al. (2016)* analyzed offline/online phase integration and noise-anomaly differentiation in novelty detection; *Barbariol et al. (2022)* evaluated tree-based methods, particularly iForest variants; *Clever et al. (2022)* constructed a structured framework for streaming classification workflows. Additionally, comprehensive reviews have addressed cross-cutting challenges in streaming data processing: *Gurjar & Chhabria (2015)* examined

concept evolution in streaming classification with methods for unknown class detection; while (*Chauhan & Shukla, 2015*) explored K-Means applications for clustering-based anomaly detection in high-volume, concept-drifting streaming data.

Notwithstanding these significant contributions of existing research, the rapid advancement of big data technologies, machine learning, and deep learning has precipitated the emergence of numerous innovative methodologies in recent years. Accordingly, this study proffers a new taxonomic framework that synthesizes both established classical methods and recently developed approaches into two primary categories: traditional machine learning and deep learning. Within the traditional machine learning paradigm, models are further classified according to their algorithmic principles: statistical models, distance models, clustering models, density models, isolation models, and frequent item mining models. Analogously, deep learning approaches are categorized as reconstruction models, generative models, predictive models, and representation learning models. Employing this structured framework, the present study undertakes a systematic review and comparative evaluation of these distinct methodological classes.

### Traditional machine learning

The methods based on traditional machine learning are as follows:

- Statistical models: These models observe and analyze observable streaming data based on principles of probability theory and statistics. They infer underlying patterns among data to detect anomalies. The category includes Fourier Transform, Wavelet Transform, Power Spectral Density, Gaussian models and Entropy models. *Hunt & Willett (2018)* used a dynamic and low-rank Gaussian mixture model for online anomaly detection in wide-area motion imagery and e-mail databases. *Tao & Michailidis (2019)* utilized higher-order statistical information to detect attackers in power systems. *Chouliaras & Sotiriadis (2019)* implemented a suite of algorithms including autoregressive integrated moving average (ARIMA), seasonal ARIMA, and long short-term memory (LSTM) to detect anomalies in sensor data. *Yu, Jibin & Jiang (2016)* leveraged ARIMA model to detect anomalies in WSN.

- Distance models: These models quantify the similarity between two sequences using explicit distances, such as Euclidean distance, Manhattan distance, Chebyshev distance and Minkowski distance. When the distance between the sequence being tested and the normal sequence exceeds the expected similarity measure range, the sequence is flagged as an anomaly. *Zhu et al. (2020)* applied min heap to compute upper bound or lower bound of distances between objects and their $k$th nearest neighbor for anomaly detection in IoT streaming. *Ma, Aminian & Kirby (2019)* employed radial basis function (RBF) to perform novelty detection and prediction on streaming data of time series. *Miao et al. (2018)* used a distributed online one-class support vector machine (OCSVM) for anomaly detection in WSN.

- Clustering models: These models map sequence data items into a $n$-dimensional space and group them into different clusters based on similarity in the latent space. If a new data item is far from the centroids of clusters or has a low probability of belonging to any

cluster, it can be considered as an anomaly. Clustering models designed for streaming data can be classified based on traditional clustering algorithms, including hierarchical-based, partition-based, density-based, grid-based, and model-based clustering. *Maimon & Rokach (2005)* provided a formal framework for understanding the key distinctions between these clustering approaches. Building on this foundation, *Mousavi, Bakar & Vakilian (2015)* demonstrated in their comprehensive study of data stream clustering algorithms that streaming data clustering differs significantly from traditional clustering in several aspects. These differences arise due to the inherent characteristics of streaming data, such as the need to read data in a specific order, processing in short time intervals, and receiving the next instance before storing the current entire stream. Recent research has extended these foundational concepts to develop more sophisticated clustering approaches for streaming data anomaly detection. *Lee & Lee (2022)* proposed a kernel-based clustering method to efficiently solve the online clustering problem of multivariate streaming data. To enhance detection accuracy in diverse streaming applications, *Degirmenci & Karal (2022)* combined local outlier factor (LOF) and density-based spatial clustering of applications with noise (DBSCAN). To contend with the challenge of noise in streaming data, *Bigdeli et al. (2018)* introduced a novel method called collective probability labeling (CPL), which combines clustering and gaussian models to gradually update clusters and mitigate the impact of noise on detection results. For large-scale streaming data processing, *Bagozi, Bianchini & De Antonellis (2021)* developed a parallelized framework for incremental clustering that achieves sustainable processing on distributed architectures. In the IoT domain, *Raut et al. (2023)* applied adaptive window and adaptive clustering techniques to infer interesting events from continuous sensor streaming data. *Bezerra et al. (2020)* tackled the fundamental problem of autonomous cluster creation and merging in streaming data using innovative online recursive clustering techniques. Table 3 synthesizes these insights to provide a contrastive analysis between clustering in streaming data and traditional clustering.

- Density models. These models determine which data points are anomalies by calculating the density around the data points. It is noteworthy that there is an overlap between density-based and distance-based models, as density-based models often rely on distance calculations. LOF is the most widely used density-based method, identifies anomalies by comparing the local neighborhood density of data points. Similarly, DBSCAN identifies noise points as anomalies through density-based clustering. Kernel density estimation (KDE) has emerged as a potent non-parametric technique for this purpose. *Zhang, Zhao & Li (2019)* utilized KDE for density estimation within sliding windows, which significantly enhanced context anomaly detection performance for streaming data. *Liu et al. (2020)* developed a top-*n* methodology based on KDE that effectively addresses local anomaly detection challenges in large-scale, high-throughput streaming environments. To overcome the limitations associated with high-dimensional data processing, more recently, *Ting et al. (2023)* demonstrated that adaptive KDE techniques can dynamically adjust to evolving data distributions, thereby providing more robust probability density estimates for anomaly assessment in non-stationary streaming

**Table 3 Comparison between streaming data clustering and traditional clustering.**

| Streaming cluster | Traditional cluster | References |
|---|---|---|
| Online processing | Offline processing | *Mousavi, Bakar & Vakilian (2015)* |
| Approximate results | Accurate results | *Lee & Lee (2022)* |
| Single-pass processing | Multi-pass processing | *Maimon & Rokach (2005)* |
| Retains essential data | All data can be stored | *Bezerra et al. (2020)* |

environments. To address the dual challenges of high-dimensional data and storage efficiency, researchers have developed various extensions and hybrid approaches. *Yang, Chen & Fan (2021)* introduced the extended LOF, which effectively solved the problems of large storage space requirements and unsatisfactory detection results for high-dimensional data. *Aggarwal & Yu (2008)* proffered a density-based methodology capable of operating effectively without assumptions regarding the underlying data distribution, thereby eliminating associated uncertainties. In the context of real-time processing, several innovative implementations have been proposed. *Shylendra et al. (2020)* implemented the KDE kernel *via* CMOS Gilbert Gaussian unit, providing a real-time statistical model for the likelihood estimation detection algorithm. *Zheng et al. (2017)* demonstrated the effectiveness of KDE for real-time outlier detection in distributed streaming data environments. The integration of density-based methods with other techniques has also yielded promising results. *Gokcesu et al. (2018)* combined density-based approaches with incremental decision tree (IDT) to construct subspaces of the observation space, effectively detecting anomalies hidden in sequential observation streaming data. *Vallim et al. (2014)* built upon the Denstream framework proposed by *Cao et al. (2006)* to develop an unsupervised automatic transformation framework based on density and entropy indicators.

- Isolation models: These models are predicated on the principle of isolating or partitioning data instances. They separate outliers from normal data points by calculating distance, similarity, or constructing boundary and hyperplane. Isolation forest (iForest) was proposed by *Liu, Ting & Zhou (2008)*, stands out as the most classic and foundational method. iForest constructed a set of isolation trees by recursively partitioning the data. Each tree isolates outliers in the shallow layers while normal points are isolated in deeper layers. Numerous enhancements have been proposed to enhance iForest for streaming data scenarios. *Shao et al. (2020)* developed AR-iForest, a combination of auto-regressive modeling and isolation forest that aims to enhance the efficacy of anomaly detection in time series data. *Heigl et al. (2021)* presented PCB-iForest, a solution to the challenges posed by high-volume, high-speed streaming data in computer networks. This implementation integrates extended iForest variants with the capacity to evaluate features based on their contribution to a sample's anomalousness.

- Frequent item mining models: Frequent item mining models mine frequently occurring patterns or items from streaming data as normal patterns. When the pattern of new data appearing in streaming data does not match these frequent patterns, it is marked as anomalous data. *Cai et al. (2020a)* proposed a two-phase minimal rare itemset mining

**Table 4 Comparison of advantages and disadvantages of traditional machine learning-based streaming data anomaly detection algorithms.**

| Model | Advantage | Disadvantage | References |
|---|---|---|---|
| Statistical Model | Capable of modeling data, inferring relationships between variables | Requires certain *a priori* assumptions, needs validation of model reliability, requires the selection of fitting data processing methods | *Hunt & Willett (2018)*, *Tao & Michailidis (2019)*, *Yu, Jibin & Jiang (2016)* |
| Distance model | Can mine data in-depth | High requirements for data preprocessing, demanding distance measurement methods, sensitive to noise | *Zhu et al. (2020)*, *Ma, Aminian & Kirby (2019)*, *Miao et al. (2018)* |
| Clustering model | Broad applicability, robust interpretability | Not suitable for high-dimensional or large-scale streaming data, sensitive to initial values, high requirements for preprocessing | *Lee & Lee (2022)*, *Raut et al. (2023)* |
| Density model | Simple to implement, quickly reveals potential structures and robust to noise | Suffers from the curse of dimensionality in high-dimensional data, computationally intensive for large-scale data | *Liu et al. (2020)*, *Zhang, Zhao & Li (2019)* |
| Isolation model | Capable of modeling data distribution, suitable for complex data distributions | Performance may decrease with high-dimensional data | *Liu, Ting & Zhou (2008)* |
| Frequent item mining | Effective at identifying outliers and anomalies in low-density areas, no need for labeled data, supports unsupervised learning | Potential for false positives due to noise and outliers in dataset | *Cai et al. (2020a)*, *Hao et al. (2019)*, *Cai et al. (2020b)* |

methodology detected anomalies in uncertain streaming data. *Cai et al. (2020b)* used min weighted rare items mining to detect anomalies in uncertain streaming data. *Hao et al. (2019)* proposed a method for mining frequent itemsets from uncertain streaming data through the construction of matrix structures and the application of upper-bound concepts.

This study organizes and categorizes the search results, revealing that the mainstream methods for streaming data research primarily include adaptations and variants of seminal algorithms, including KNN, iForest, ARIMA, and LOF. Furthermore, a significant body of work is devoted to hybrid approaches that synergistically combine multiple models. Table 4 below summarizes the respective strengths and limitations of streaming data anomaly detection algorithms based on traditional machine learning.

### Deep learning

Deep learning models are also crucial in streaming data analysis.

- Reconstruction models: They detect anomalies by learning the reconstruction error of the input data. This process involves encoding the input data into a lower-dimensional latent space and then decoding this representation back into the original data space. Anomalies are identified by comparing the differences between the original and the reconstructed data. *Yoo, Kim & Kim (2019)* utilized a recurrent reconstruction network (RRN) for anomaly detection in temporal data. *Zeng et al. (2023a)* employed a stacked autoencoder (AE) to better distinguish between nuanced anomalies and subsequently enhanced detection accuracy through a joint optimization with KDE.
- Generative models: Within a comprehensive taxonomic framework, generative paradigms constitute a sophisticated class of detection methodologies that transcend

conventional pattern recognition. These models operate by cultivating the capacity to synthesize artificial data instances that mirror authentic distributions, subsequently facilitating anomaly identification through comparative analysis between authentic and synthetic data manifestations. Particularly prominent within this classification are variants of the Boltzmann machine and the generative adversarial network (GAN) architecture. *Xing, Demertzis & Yang (2020)* orchestrated a pioneering implementation of real-time evolving peak-constrained Boltzmann machine for anomaly detection within IoT, demonstrating how such approaches can be seamlessly integrated into the framework for real-time streaming analytics. Advancing this trajectory, *Talapula et al. (2023)* engineered an intricate fusion of search and rescue brain-storm optimization (SAR-BSO) with hybrid feature selection (FS) and deep belief network (DBN) classifiers, establishing a multilayered approach for the identification and localization of anomalous patterns within streaming log environments. The GAN architectural paradigm has undergone extensive refinement for anomaly detection, marked by pioneering contributions. *Li et al. (2019)* adeptly utilized long short-term memory-recurrent neural network (LSTM-RNN) frameworks to encapsulate intricate multivariate spatiotemporal interdependencies in cyber-physical systems. *Hallaji, Razavi-Far & Saif (2022)* ingeniously integrated dynamic temporal attributes of streaming data into GAN-based detection modules, significantly enhancing intrusion detection capabilities in Internet of Things (IoT) ecosystems. *Grekov & Sychugov (2022)* proposed sophisticated distributed processing paradigms, leveraging GAN architectures to synthesize realistic network traffic, thereby augmenting detection precision while concurrently alleviating computational demands in the analysis of voluminous network packets.

- Prediction models: They primarily learn the intricate relationships between input data and target variables, formulating a sophisticated function approximation model. Anomalies are identified by comparing the differences between predicted and actual values. RNN and LSTM are frequently adopted as the cornerstone architectures for these models. *Wang et al. (2023)* developed an enhanced LSTM-AE to detect runtime reliability anomalies in MEC services based on distributional discrepancy evaluation. *Liu et al. (2021)* employed both standard and enhanced LSTM for the real-time monitoring and correction of aberrant data in IoT. *Cheng et al. (2019)* proposed a semi-supervised hierarchical stacked temporal convolutional network (TCN) to facilitate anomaly detection in smart home communication.

- Representation models: They employ multi-layer neural networks to learn abstract feature representations, thereby capturing complex patterns and anomalous behaviors. Common models in this category utilize convolutional neural network (CNN) or graph neural network (GNN) as their underlying structures. *Munir et al. (2018)* used CNN to detect common periodic and seasonal outlier anomalies in streaming data. *Garg et al. (2019)* proposed a hybrid method based on grey wolf optimization (GWO) and CNN for anomaly detection in network traffic of cloud data centers.

Drawing upon recent investigations, this study systematically organizes and categorizes prevailing research methodologies for streaming data anomaly detection. These methods

**Table 5 Comparison of advantages and disadvantages of streaming data anomaly detection algorithm based on deep learning model.**

| Model | Advantage | Disadvantage | References |
|---|---|---|---|
| Reconstruction model | No need for labeled anomaly samples, can capture local and global features of data | Higher false-positive rate for high-dimensional and large-scale data, requires a large amount of training data to learn data distribution and patterns | *Yoo, Kim & Kim (2019)*, *Zeng et al. (2023a)*, *Xu et al. (2023)* |
| Generative model | Can model complex, high-dimensional data distributions, can learn data distribution to generate new samples, no need for labeled data | Training and inference processes are complex and time-consuming for complex data distributions and high-dimensional data, prone to mode collapse which can result in a lack of diversity in generated samples | *Xing, Demertzis & Yang (2020)*, *Talapula et al. (2023)*, *Li et al. (2019)* |
| Predictive model | Can capture dynamic changes and trends in data, excellent detection performance for time-series data | Issues with gradient vanishing and exploding, need to continuously adapt to new data distributions for non-stationary data | *Wang et al. (2023)*, *Liu et al. (2021)* |
| Representation learning model | Suitable for high-dimensional, complex, and large-scale data, better understanding of the intrinsic structure and features of data, can automatically extract useful features | Requires a large amount of training data and computational resources to train deep neural networks, may have poor interpretability in some cases, prone to overfitting | *Munir et al. (2018)*, *Garg et al. (2019)* |

include using AE, VAE, GAN, RNN, CNN, and LSTM as the foundational structures, the integration of diverse model combinations, and advanced deep learning paradigms, including reinforcement learning (*Zhou, Zhang & Hong, 2019*), transfer learning (*Wang et al., 2021*). Table 5 summarizes the strengths and limitations of deep learning-based streaming data anomaly detection algorithms.

## Anomaly and output type

To facilitate systematic analysis and treatment of anomalies, it is essential to establish a comprehensive classification method that encompasses both anomaly types and output types.

- **Classification of anomaly types:** Anomaly values are typically classified into three types: point anomaly, contextual anomaly, and collective anomaly (*Gorunescu, 2011*). Point anomaly refers to isolated data points that are markedly different from other data points. Contextual anomaly are data points that deviate from normal behavior or patterns within a specific context compared to other data points in given context. Collective anomaly refers to anomalies relative to the entire dataset.

- **Classification of output types:** Output result types are generally divided into label and score (*Chandola, Banerjee & Kumar, 2009*). Anomaly labels allow for direct determination of whether each point is an anomaly based on the model's output. Anomaly scores provide further insight into which points exhibit a higher degree of anomaly.

## STUDY OF LITERATURE AND DISCUSSIONS

In the above discussions, this investigation delineated the datasets utilized for detection purposes, the algorithms for measurement methodologies, the detection algorithms, and

the types of anomaly. Subsequently, this study will concentrate on the methodologies for measuring streaming data. This study will conduct a theoretical appraisal of literature, guided by a suite of formulated evaluative criteria, culminating in an extensive discourse.

## Evaluation criteria

This research proposes a comprehensive set of evaluation criteria for assessing existing research. As shown in Table 6, these criteria encompass a broad spectrum of research outcomes. Specifically, the evaluation criteria mainly include efficiency metrics, evaluation metrics, task metrics, and anomaly interpretability.

- **Efficiency metrics:** Efficiency metrics serve to quantify the efficiency of algorithm, particularly with respect to their algorithmic complexity.
- **Evaluation metrics:** Evaluation metrics are utilized to evaluate the efficacy of algorithm. They primarily encompass the following: false alarm rate (FAR), recall, detection rate (DR), true positive rate (TPR), true negative rate (TNR), receiver operating characteristic curve (ROC), and area under curve (AUC). In addition to the above evaluation metrics, additional indices, such as the Youden's index (*Harush, Meidan & Shabtai, 2021*) and the Kappa coefficient (*Xing, Demertzis & Yang, 2020*; *Jain, Kaur & Saxena, 2022*) are also used as reference standards.
- **Task metrics:** Task metrics assess algorithm's capability in managing concept drift and feature drift, as well as its aptitude for handling high-dimensional or large-scale streaming datasets.
- **Anomaly explanation:** Anomaly explanation indicates whether algorithm can provide explanations.

## Detection algorithm discussion

The vast majority of traditional machine learning models are lightweight, meaning they have compact model size and low memory requirements. Therefore, these algorithms are widely used in resource-constrained environments.

Statistical models, valued for their parsimony and straightforward implementation, have demonstrated considerable utility across a multitude of applications. *Hunt & Willett (2018)* adapted gaussian mixture model for online anomaly detection in wide-area motion imagery and email databases. *Tao & Michailidis (2019)* used higher-order statistical techniques to detect false data injection (FDI) attacks in power systems. *Ma, Aminian & Kirby (2019)* applied ARIMA to detect anomalous traffic in WSN. Nevertheless, a principal constraint of such statistical methodologies is their inherent reliance on specific distributional assumptions. This prerequisite can substantially limit their effectiveness when confronted with the complexity of real-world data, where such assumptions may not be valid.

Distance models facilitate localization and identification of anomalies by leveraging distance or similarity measures. *Zhu et al. (2020)* employed min heap to compute upper bounds or lower bounds of distances between objects and their $k$-th nearest neighbors for

**Table 6 Literature evaluation of streaming data detection in the field of network security.**

| Ref. | AD | CD | EC | DP | Explain | Effectiveness | Output | Type |
|---|---|---|---|---|---|---|---|---|
| *Bhatia et al. (2022)* | Statistical Model | * | AUC, ROC, Acc, Recall | Batch | ✗ | Scalable | Score | Point |
| *Tong & Prasanna (2017)* | FIM | * | Precision, Recall | Stream | ✗ | Adaptive | Score | Point |
| *Hao et al. (2019)* | Cluster, LSTM, AE | ✓ | Deland, FPR, ROC-AUC, Recall | Stream | ✗ | Robust | Cluster label | Contextual |
| *Hoeltgebaum, Adams & Fernandes (2021)* | Statistical Model | ✓ | FD, FP, FN, MSE, MAE | Stream | ✗ | Adaptive | Label | Point |
| *Nadler, Aminov & Shabtai (2019)* | iForest | * | DR, FPR | Batch | ✓ | Robust | Score | Collective |
| *Wambura, Huang & Li (2022)* | RNN | * | MAE, ROC-AUC | Stream | ✓ | Scalable, robust | Score | * |
| *Xiaolan et al. (2022)* | Cluster | ✓ | DR, FAR, Acc | Stream | ✗ | Robust | No | Collective |
| *Yin, Li & Yin (2020)* | Statistical Model | * | EDR, EFP, END, ENF, EFR | * | ✗ | Robust | Score | Point |
| *Zeng et al. (2023b)* | Bloom Filter | * | DR, FAR | Stream | ✗ | Robust | Label | Point |
| *Wahab (2022)* | DNN | ✓ | Precision, Recall, F1, TP, FP, FN, TN, Acc | * | ✗ | Robust | Label | Collective |
| *Cai et al. (2022)* | K-Means, Cluster | ✓ | Acc | Stream | ✗ | Robust | Label | All |
| *Cheng et al. (2020)* | TCN | * | Acc, Precision, Recall, F1 | Batch | * | Robust | Label | Collective |
| *Jain, Kaur & Saxena (2022)* | K-Means, Cluster, SVM | ✓ | Acc, FAR, Precision, Recall, F1, Kappa Statistic | Stream | * | Adaptive, robust | Label | Collective |
| *Mirsky et al. (2017)* | Cluster | ✓ | ROC, AUC, TPR, FPR | Stream | * | Robust | Score | All |
| *Scaranti et al. (2022)* | DBSCAN, Entropy | ✓ | Acc, Precision, Recall, F-measure, FAR | Stream | ✗ | Adaptive | Label | All |
| *Shao et al. (2023)* | Bloom filter | ✓ | Acc | Stream | * | Robust | Label | All |
| *Xing, Demertzis & Yang (2020)* | e-SNN, REBOM | ✓ | K-Stats, K-Temp-Stats | Stream | * | Robust, scalable | Label | Point |
| *Xu et al. (2023)* | AE, SVM | ✓ | AUC | Stream | ✗ | Robust | Score | Point |
| *Yang et al. (2021)* | XGboost | ✓ | AUC | Stream | ✗ | | Score | Point |
| *Zeng et al. (2023a)* | KDE, AE | ✓ | Recall, Precision, F-score, ROC, TPR, FPR, AUC | Stream | ✗ | Scalable, adaptive, robust | Score | Point |
| *Zhou et al. (2020)* | Variational LSTM | * | Precision, Recall, F1, FAR, AUC | Batch | ✗ | Scalable | Label | Point |
| *Saheed, Abdulganiyu & Tchakoucht (2023)* | GWO, ELM, PCA | * | Precision, Recall, DR, Acc | Batch | * | Scalable, robust | Label | Point |
| *Yoon et al. (2022)* | AE | ✓ | AUC | Batch | ✗ | Scalable, adaptive, robust | Score | Point |
| *Yu et al. (2018)* | Cluster | * | AUC, Acc | Stream | ✗ | Scalable, adaptive, robust | Score | Point |

**Note:**
AD, anomaly detection; CD, concept drift; EC, evaluation criteria, DP, data processing mode; FIM, frequency itemset mining; FD, false detection; MAS, mean average score; EDR, event detection rate; EFP, event false positive rate; END, error node detection rate; ENF, error node false positive rate; EFR, error node false recognition rate; *, unknown; ✗, not support; ✓, support.

anomaly detection in IoT. *Miao et al. (2018)* implemented online OCSVM in distributed WSN. However, distance-based models often face the curse of dimensionality and high computational costs when dealing with high dimensional or large-scale data. It is essential to adopt suitable dimensionality reduction methods and optimize distance metrics to ease these challenges.

Cluster models adeptly adapt to intrinsic structural dynamics, rendering them well-suited for addressing distributional shifts in streaming data. *Jain, Kaur & Saxena (2022)*, *ZareMoodi, Kamali Siahroudi & Beigy (2019)* used cluster-based models to address concept drift. *Wang et al. (2022)* analyzed the influence of collective anomalies based on cluster. *Zou et al. (2023)* combined grid cluster with gaussian model to improve the ability to distinguish noise from anomalies. *Harush, Meidan & Shabtai (2021)* integrated cluster-based methods with deep learning to classify contextual information in real time. *Bah et al. (2019)* employed micro-clusters to refine the search space for outlier detection. *Wang et al. (2020b)* utilized the centers of micro-cluster within each class as inputs for detecting unknown classes by projecting micro-cluster centers onto fixed positions on orthogonal axes in the feature space, forming clear classification boundaries.

Density models effectively address local anomaly detection in high-dimensional or large-scale streaming data with with reduced memory requirements. *Liu et al. (2020)* introduced a KDE-based outlier detection method that substantially accelerates processing speed through an upper-bound pruning strategy. *Chen, Wang & Yang (2021)* employed entropy-weighted index calculation and reachable distance factor discrimination methods, achieving a 15% improvement in accuracy while requiring only 1% of the runtime compared to the traditional LOF. Density-based models can adapt to the evolution of streaming data, but they have high computational complexity and are challenging to apply to high-dimensional sparse or large-scale streaming data.

Isolation models exhibit robust real-time capability (*Shao et al., 2020*). However, they are characterized by high computational complexity and sensitive to model parameters. Consequently, the development of more efficient isolation frameworks is imperative to improve detection speed.

Frequency item mining algorithms can uncover common patterns in streaming data, thereby enhancing the recognition and pinpointing of anomalous samples. *Cai et al. (2020a, 2020b)*, *Hao et al. (2019)* stored uncertain streaming data information in matrices to detect outliers. Compared to conventional static data frequent item mining algorithms, these approaches are more suitable for processing large-scale uncertain streaming data. Owing to its ability to handle complex and large-scale data, automatically learn relevant features from streaming data, and provide automatic feature extraction, scalability, adaptability and handling of complex relationships in streaming data, deep learning-based algorithms are becoming increasingly popular.

Reconstruction models effectively learn high-level feature representations of data without the need for manual feature design. These models provide reconstruction errors or losses for anomalous data, making the models and results easier to understand and interpret. *Yoo, Kim & Kim (2019)*, *Zeng et al. (2023a)* used AE to detect anomalies by analyzing the reconstruction errors between the reconstructed and original sequences.

Nevertheless, reconstruction models have high complexity and require enough normal data samples as training sets to establish the distribution model for normal data.

Generative models not only capture complex streaming data distributions but also generate new samples based on the learned distributions. The GAN architecture stands as a paradigmatic exemplar in this domain, demonstrating remarkable efficacy in anomaly detection. *Li et al. (2019)* advanced the conventional GAN framework by integrating LSTM-RNN as its foundational architecture, thereby exploiting the intricate spatiotemporal correlations and multivariate dependencies inherent in sequential data streams. Extending this trajectory of innovation, *Hallaji, Razavi-Far & Saif (2022)* engineered an advanced approach that incorporates the dynamic temporal characteristics of streaming data directly into the GAN detector framework, yielding substantial improvements in intrusion detection precision within IoT ecosystems. In conjunction with GAN-based methodologies, restricted Boltzmann machines (RBMs) represent an equally viable analytical tool within the proposed classification paradigm. *Xing, Demertzis & Yang (2020)* pioneered enhancements in anomaly detection fidelity through the introduction of real-time evolving spiking RBM architectures, which demonstrate particular aptitude for processing high-velocity data streams. Further advancing this architectural lineage, *Talapula et al. (2023)* orchestrated a sophisticated fusion of RBMs with deep belief criteria and metaheuristic algorithms, specifically calibrated for generative streaming log data analysis, thereby substantially augmenting the model's discriminative capabilities for anomaly identification.

Prediction models with their powerful expressive capabilities and adaptability can capture temporal dependencies and patterns. *Wang et al. (2023)* combined LSTM prediction models with AE to compress time series data into low-dimensional feature models. Compared to employing a single prediction-based model, combining with other types of deep learning models improves the efficiency and effectiveness of real-time streaming data detection. *Liu et al. (2021)* used LSTM+ for real-time monitoring and correction of streaming data in IoT. Similarly, *Cheng et al. (2019)* stacked distance-based models such as KNN and SVM, with probabilistic models, includeing decision trees and Bayesian classifiers and combined them with TCN for sequence problems.

Representation models can capture the underlying features and intrinsic structures of streaming data. *Munir et al. (2018)* used a CNN-based representation learning model, termed DeepAnT, to address periodic and seasonal anomalies that cannot be solved by distance-based and density-based anomaly detection techniques in traditional machine learning. *Garg et al. (2019)* improved the standard CNN employing a uniform distribution method for anomaly detection in heterogeneous streaming data.

## Measurement algorithm discussion

Frequency estimation can estimate the occurrence frequency of various elements in streaming data. Consistent hashing and counters can efficiently estimate element frequencies in a single pass, characterized by minimal time and space complexity. These approaches yield error bounds for frequency approximation, meeting the real-time

detection requirements. However, there is a certain estimation error compared to exact calculations due to the use of approximate counters.

Quantile estimation elucidates the global distribution of streaming data by calculating the approximate quantiles in real time. However, in cases where the data distribution in streaming is extremely uneven, the estimation of tail quantiles may be less accurate (*Liu et al., 2018*).

Change detection employs chi-square tests or the KL divergence to detect the statistical distribution and aggregate metrics of streaming data. However, this approach is sensitive to parameter settings (*Kuncheva, 2011*).

Cardinality estimation leverages probabilistic data structures to efficiently estimate the cardinality of streaming data in a single pass. However, compared to frequency estimation, real-time performance may be inferior (*Jie et al., 2022*).

Similarity calculation utilizes LSH or other algorithms to rapidly retrieve neighbors and compute approximate similarities in sublinear time in streaming data. However, similarity calculations for high-dimensional and large-scale streaming data may be time-consuming, requiring appropriate similarity measurement methodologies and thresholds (*Wu et al., 2024*).

Sampling techniques represent a critical dimension in the study taxonomy of streaming data anomaly detection methods, as they extract representative subsets from streaming data, significantly reducing storage and processing requirements. While effective sampling demands meticulously designed random or stratified strategies, sketch algorithms have emerged as particularly promising within this category due to their sublinear complexity and efficiency advantages over traditional item-by-item processing approaches. Recent advancements in sketch algorithms demonstrate their increasing relevance to the framework of streaming anomaly detection. *Liu et al. (2016)* developed a universal monitoring framework that exemplifies how sketching can provide system-wide visibility while preserving computational efficiency. Building on this foundation, *Yang et al. (2018)* addressed the challenge of varying traffic conditions through adaptive sketching methods, a critical capability for detecting anomalies in dynamically evolving streams. To address the need for scalable detection in high-throughput network scenarios, *Huang et al. (2017)* proposed a robust measurement framework. The evolution of sketch-based algorithms has focused primarily on two aspects central to effective anomaly detection: accuracy and performance. *Liu et al. (2015)* tackled the fundamental challenge of hash collisions by reconstructing and estimating infected host cardinality using overlapping techniques of hash bit strings based on vectorized Bloom Filter. This approach significantly enhances detection accuracy, addressing one of the key challenges identified in the study framework. More recently, *Zeng et al. (2023b)* advanced this concept by establishing a multi-layer dLSHBF model using Bloom Filter, which effectively avoids element conflicts by reducing data hash encoding length. Similarly, *Xiao et al. (2023)* employed a multi-level design methodology combined with TailCut's register compression technique (*Xiao et al., 2020*) to alleviate hash collisions between streams, demonstrating how hash function selection critically impacts detection algorithm performance. Within this study comprehensive framework of streaming data anomaly detection methodologies, these sketch-based

approaches represent a particularly valuable direction for network security applications, offering an optimal balance between computational efficiency and detection accuracy.

# FUTURE DIRECTIONS AND OPEN RESEARCH CHALLENGES

Although many anomaly detection algorithms for streaming data have been proposed, there are still some challenges and limitations that make it impractical to use these algorithms in the real world. Therefore, in this section, the study will describe the limitations of existing research literature and propose research questions and suggestions in the context of network security.

## Limitations of current research

This study surveys the latest literature on anomaly detection in streaming data and discusses some issues in the research direction of streaming data in the field of network security. During the research process, this study verified that the existing literature in this field has not adequately addressed the relevant problems, thus requiring more attention. Based on the previous statements and research, the following general limitations of existing research literature can be identified:

1. The field of cybersecurity has not adequately addressed the issue of high-dimensional streaming data.
2. The interpretability of detected anomalies in streaming data is poor.
3. There is still significant room for improvement in the efficiency of these models or methods.
4. There is a lack of methods for handling anomalies in multi-type data.
5. There is a scarcity of datasets appropriate for streaming data in the field of network security.

## Future research directions

**Anomaly detection in high-dimensional streaming data:** Anomaly detection in high-dimensional streaming data typically requires a significant amount of feature engineering. Deep learning can automatically learn relevant features, with its multi-layer structure, proficiently manage high-dimensional streaming data enhance its effectiveness by learning more abstract representations through the hierarchical extraction and combination of features. However, the application of deep learning algorithms in the field of cybersecurity, especially in high-dimensional streaming data, has not been extensively studied. Therefore, applying deep learning to anomaly detection in streaming data is a direction worth exploring. It is worth noting that high-dimensional streaming data requires more hidden layers for learning input features, resulting in a linear increase in model computational complexity with the increase of hidden layers.

**Detection of anomalies in multi-type data:** Different applications generate diverse streaming data, such as text streaming, image streaming, video streaming, *etc*. Detecting

anomalies in these heterogeneous streaming data can be cumbersome. Therefore, there is a need to research new algorithms or techniques to handle these complex data formats to better understand streaming data in the real world.

**Interpretability of anomaly:** Recent studies have underscored the significance of anomaly interpretation, particularly in the context of streaming data. The aim is to unearth plausible explanations for detected abnormal patterns. This will help in understanding and evaluating relevant detection results and enhance the reliability of anomaly evaluation. So far, the existing literature has focused on interpreting anomalies in low-dimensional streaming data, thus requiring further investigation in this area.

**Improvement of detection algorithm efficiency:** The characteristics of streaming data require anomaly detection algorithms to produce results at a lower computational cost, making the efficiency of algorithms crucial. Due to limitations in time and storage space, it is worth considering compressing the streaming data first and then combining traditional machine learning algorithms or deep learning models for anomaly detection. This approach can improve efficiency while ensuring detection accuracy.

**Application of large language models:** Large language models (LLMs) offer a transformative approach to anomaly detection in network security streams by enabling the analysis of textual and semi-structured data at a deep semantic level. By processing diverse data sources, such as network logs, system alerts, or command-line sequences, LLMs can harness their sophisticated contextual comprehension to detect subtle and intricate anomalies that often elude conventional detection methodologies. For example, they are capable of identifying emerging phishing campaigns or multi-stage insider threats by analyzing the narrative flow and contextual coherence of communication streams. Despite this immense potential, their practical application remains an underexplored research area, primarily due to significant challenges. The high computational requirements and inference latency of LLMs pose a barrier to real-time analysis, while the need for curated, domain-specific datasets for effective fine-tuning presents another obstacle. Therefore, future research should be strategically directed towards enhancing the viability of LLMs for this task. Investigating lightweight architectures, such as distilled or pruned models, and exploring sophisticated transfer learning methodologies represent promising pathways to harness the power of LLMs for real-time, adaptive anomaly detection in cybersecurity.

## CONCLUSIONS

This survey provides a comprehensive overview of streaming data anomaly detection in network security, systematically categorizing existing research based on datasets, measurement techniques, detection algorithms, anomaly types, and output types. Through this categorization, evaluation criteria have been derived to assess the characteristics, advantages, and limitations of various research approaches. It has been established that streaming data anomaly detection is critical in addressing the challenges posed by the increasing volume and complexity of data in cybersecurity applications. Several key challenges have been identified, including the need for efficient algorithms to handle

high-dimensional and multi-type data, improved interpretability of anomalies, and the exploration of emerging techniques such as LLMs. It is anticipated that this survey will enhance understanding of this vital research area and provide valuable guidance for future investigations. Future research is expected to focus on two primary directions. First, a more in-depth analysis of mainstream literature is planned to offer scholars and practitioners a thorough understanding of current research developments. Second, it is proposed to develop a novel anomaly detection method to address the limitations observed in existing approaches, leveraging the unique characteristics of streaming data to improve efficiency. Specifically, an efficient anomaly detection algorithm will be designed and its performance and accuracy validated through rigorous experiments.

### Funding

The author received no funding for this work.

### Competing Interests

The author declares that they have no competing interests.

### Author Contributions

- Pengju Zhou conceived and designed the experiments, performed the experiments, analyzed the data, performed the computation work, prepared figures and/or tables, authored or reviewed drafts of the article, and approved the final draft.

### Data Availability

This is a literature review.

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
