# Peer review of "A survey of streaming data anomaly detection in network security"

_PeerJ Computer Science, doi:10.7717/peerj-cs.3066_

## Round 0.1 · original submission · Major Revisions

Reviewers have now commented on your paper. The reviewers have raised concerns regarding the analyzed papers, research gaps, previously reported surveys, and contributions. These issues require a major revision. Please refer to the reviewers’ comments at the end of this letter; you will see that they advise you to revise your manuscript. If you are prepared to undertake the work required, I would be pleased to reconsider my decision. Please submit a list of changes or a rebuttal against each concern when you submit your revised manuscript.

Thank you for considering PeerJ Computer Science for the publication of your research.

With kind regards,

Reviewer 1 ·

Basic reporting

A survey of streaming data anomaly detection in network security

This paper presents a survey of 42 articles on streaming data anomaly detection in cybersecurity and categorizes the scenarios, data types, research methods and problems addressed. This research primarily achieves two objectives: (1) systematically categorize existing research to help network security researchers better understand the current research. (2) summarize several future research directions based on the classification results to help researchers quickly focus on these directions.

Experimental design

no comment

Validity of the findings

Remarks

1. The paper takes into account 42 papers related to the topic of streaming data anomaly detection in network security and proposes the following classification: dataset, measurement, detection algorithms, anomaly type and output type. This classification is described in section 4.

2. However, on one hand, there are subsections in section 4 without any reference, for example subsections 4.1 and 4.4. The author should explain which papers belong to this classification. The same remark for the measurement technique Quantile estimation.

On the other hand, there are many references that requiere further explanation or to relate them to the sentence that procedes them. For example section 4.3:
a) Lines 196-198: Researchers have conducted comprehensive studies on a variety of methods aimed at detecting anomalies within streaming data {for example, such as, etc.} Wang et al. (2019); Boukerche et al. (2020); Din et al. (2021); Bhaya and Alasadi (2016); Souiden et al. (2017).
b) Lines 198-200: Some researchers focus on specific application scenarios {which ones?} Fahy et al. (2022), Stahmann and Rieger (2021) or specific types of methods {which ones?} Krawczyk et al. (2017); Faria et al. (2016); Barbariol et al. (2022); Clever et al. (2022).
c) Lines 204-205: There are also comprehensive survey works available {in what?} Gurjar and Chhabria (2015); Chauhan and Shukla (2015).
d) Lines 240-242: Table 3 supplies contrastive analysis between clustering in streaming data and traditional clustering Mousavi et al. (2015); Maimon and Rokach (2005). ¿Table 3 was taken from these references or the author should explain why the references appear in the sentence?
e) Lines 255-256: Kernel Density Estimation (KDE) assesses anomalies by calculating the data’s probability density estimate {such as it was proposed in?} Ting et al. (2023).
f) Lines 319-321: These methods include using AE, VAE, GAN, RNN, CNN, LSTM as the underlying structures, combining different models, and other deep learning models Zhou et al. (2019); Wang et al. (2021).
The author should explain why these references appear in the sentence.
3. Tables 2, 3, 4, and 5 requiere references.

Additional comments

Minimal remarks

1. Insert spaces between words
a) Lines: 26, 27, 28, 29, 73, 243, 397, 407, 408, 442, 447
2. Write capital letter after dot
a) Lines: 182, 190, 225, 272
3. Write capital letter in (IoT) and IEEE
a) Lines: 557, 673, 634, 656,
4. Reference Stahmann, and Rieger (2021) is incomplete

Cite this review as

Reviewer 2 ·

Basic reporting

no comment

Experimental design

no comment

Validity of the findings

no comment

Additional comments

In the abstract, the author says " However, network anomaly detection faces significant challenges when dealing with massive traffic, logs, and other forms of streaming data". What is technique implemented in this research to solve this issue.
It is a survey paper. How the survey is conducted and what is the reason for conducting this survey is not mentioned?
Already many surveys are available for the anomaly detection. What is the difference between the current research and previous researches?
The language of the paper must be improved. There are many grammatical and typo errors are in the paper.
The research gaps identified in the existing methods are to be listed.
The paper is containing unnecessary section headings. Check the standard publications and make the section headings similar to that.
In what basis this survey is conducted? How many papers are considered in this survey?
The survey of the existing methods are not given? What are techniques are already developed.
The title and content of the paper is not matching. The author says this work is survey. However, the results of the existing methods, advantages and disadvantages of the models are not analyzed properly.
The limitations and future work of the research are not given.

Cite this review as

---

## Round 0.2 · Minor Revisions

All concerns raised by the reviewers have been addressed satisfactorily; however, the paper still needs further work regarding English grammar and experimental results. These issues require a minor revision. If you are prepared to undertake the work required, I would be pleased to reconsider my decision. Please submit a list of changes or a rebuttal against each point that is being raised when you submit your revised manuscript.

**Language Note:** The Academic Editor has identified that the English language must be improved. PeerJ can provide language editing services - please contact us at [email protected] for pricing (be sure to provide your manuscript number and title). Alternatively, you should make your own arrangements to improve the language quality and provide details in your response letter. – PeerJ Staff

Reviewer 1 ·

Basic reporting

A survey of streaming data anomaly detection in network security

This paper presents a survey of 42 articles on streaming data anomaly detection in cybersecurity and categorizes the scenarios, data types, research methods and problems addressed. This research primarily achieves two objectives: (1) systematically categorize existing research to help network security researchers better understand the current research. (2) summarize several future research directions based on the classification results to help researchers quickly focus on these directions.

Experimental design

no comment

Validity of the findings

no comment

Additional comments

The author has taken into account the remarks of the first review, however the following observations are recommended:
a) The author should propose in section 7.2 the usage of large language models (LLM) in streaming data anomaly detection in network security and not only deep learning and machine learning algorithms.

b) Avoid using personal pronouns such as "I hope," "I have observed," "I will design" in the conclusions section and instead use indefinite pronouns such as "it is expected, it has been observed," "it will be proposed," "it will be designed," etc.

Cite this review as

---

## Round 0.3 · accepted · Accept

I am pleased to inform you that your work has now been accepted for publication in PeerJ Computer Science.

Please be advised that you cannot add or remove authors or references post-acceptance, regardless of the reviewers' request(s).

Thank you for submitting your work to this journal. I look forward to your continued contributions on behalf of the Editors of PeerJ Computer Science.

With kind regards,

Reviewer 1 ·

Basic reporting

A survey of streaming data anomaly detection in network security

This paper presents a survey of 42 articles on streaming data anomaly detection in cybersecurity and categorizes the scenarios, data types, research methods and problems addressed. This research primarily achieves two objectives: (1) systematically categorize existing research to help network security researchers better understand the current research. (2) summarize several future research directions based on the classification results to help researchers quickly focus on these directions.

Experimental design

no comment

Validity of the findings

The author has taken into account the remarks of the previous reviews

Additional comments

The author has taken into account the remarks of the previous reviews

Cite this review as